# Partial Pancreatic Parenchymal Atrophy Is a New Specific Finding to Diagnose Small Pancreatic Cancer (≤10 mm) Including Carcinoma in Situ: Comparison with Localized Benign Main Pancreatic Duct Stenosis Patients

**DOI:** 10.3390/diagnostics10070445

**Published:** 2020-07-01

**Authors:** Kentaro Yamao, Mamoru Takenaka, Rei Ishikawa, Ayana Okamoto, Tomohiro Yamazaki, Atsushi Nakai, Shunsuke Omoto, Ken Kamata, Kosuke Minaga, Ippei Matsumoto, Yoshifumi Takeyama, Isao Numoto, Masakatsu Tsurusaki, Takaaki Chikugo, Yasutaka Chiba, Tomohiro Watanabe, Masatoshi Kudo

**Affiliations:** 1Department of Gastroenterology and Hepatology, Kindai University Faculty of Medicine, Osaka-Sayama, Osaka 589-8511, Japan; mamoxyo45@gmail.com (M.T.); rei_i_0419@yahoo.co.jp (R.I.); a-o-k@mail.goo.ne.jp (A.O.); chochiko.4kg@gmail.com (T.Y.); nakai_agmc@yahoo.co.jp (A.N.); shunsuke.oomoto@gmail.com (S.O.); ky11@leto.eonet.ne.jp (K.K.); kousukeminaga@yahoo.co.jp (K.M.); tomohiro@med.kindai.ac.jp (T.W.); m-kudo@med.kindai.ac.jp (M.K.); 2Department of Surgery, Kindai University Faculty of Medicine, Osaka-Sayama, Osaka 589-8511, Japan; ippeimm@gmail.com (I.M.); takeyama@surg.med.kindai.ac.jp (Y.T.); 3Department of Diagnostic Radiology, Kindai University Faculty of Medicine, Osaka-Sayama, Osaka 589-8511, Japan; numoto.isao@hotmail.co.jp (I.N.); mtsuru@dk2.so-net.ne.jp (M.T.); 4Department of Pathology, Kindai University Faculty of Medicine, Osaka-Sayama, Osaka 589-8511, Japan; tchikugo@mac.com; 5Clinical Research Center, Kindai University, Osaka–Sayama, Osaka 589-8511, Japan; chibay@med.kindai.ac.jp

**Keywords:** carcinoma in situ, computed tomography, early diagnosis, pancreatic parenchymal atrophy, small pancreatic cancer

## Abstract

Background: This study aimed to evaluate and identify the specific CT findings by focusing on abnormalities in the main pancreatic duct (MPD) and pancreatic parenchyma in patients with small pancreatic cancer (PC) including carcinoma in situ (CIS). Methods: Nine CT findings indicating abnormalities of MPD and pancreatic parenchyma were selected as candidate findings for the presence of small PC ≤ 10 mm. The proportions of patients positive for each finding were compared between small PC and benign MPD stenosis groups. Interobserver agreement between two independent image reviewers was evaluated using kappa statistics. Results: The final analysis included 24 patients with small PC (including 11 CIS patients) and 28 patients with benign MPD stenosis. The proportion of patients exhibiting partial pancreatic parenchymal atrophy (PPA) corresponding to the distribution of MPD stenosis (45.8% vs. 7.1%, *p* < 0.01), upstream PPA arising from the site of MPD stenosis (33.3% vs. 3.6%, *p* = 0.01), and MPD abrupt stenosis (45.8% vs. 14.3%, *p* = 0.03) was significantly higher in the small PC group than in the benign MPD stenosis group. Conclusions: The presence of partial PPA, upstream PPA, and MPD abrupt stenosis on a CT image was highly suggestive of the presence of small PCs including CIS.

## 1. Introduction

Pancreatic cancer (PC) has the worst survival rate among the common types of cancer as shown by a low three-year survival rate (3%) in unresectable PC [1]. Although various diagnostic and therapeutic approaches have been employed to improve the prognosis of PC, remarkable success has not been achieved yet. However, the 10-year survival rates in PC patients for stage 0, stage I (TS1a), and stage I (TS1b) PC are 94.7%, 93.8%, and 78.9%, respectively [2]. Thus, the prognosis of PC patients diagnosed at stages 0 or I is extremely good, which clearly shows the importance of diagnosing PC at very early stages.

Computed tomography (CT), magnetic resonance imaging (MRI), and endoscopic ultrasonography (EUS) are widely used for the imaging diagnosis of PC [3]. Although CT and MRI are useful for the objective detection of advanced PC, these modalities are not sufficient for the diagnosis of small PC. Different from CT and MRI, EUS is powerful detection tool for PC ≤ 20 mm in size [4,5]. However, EUS is not a general imaging modality available in every hospital and its findings are highly dependent on the skill and experience of the examiner. Therefore, there is an urgent need for the identification and establishment of diagnostic findings for early stage PC that are obtained in a more general imaging modality such as CT.

Previous studies attempted to establish secondary CT findings specific for advanced and early PCs in terms of abnormality in main pancreatic duct (MPD) and parenchymal structures [6,7,8,9]. Localized MPD stenosis with or without upstream MPD dilation is generally observed in PC [10]. Localized MPD stenosis is often detected even in early PC, but this finding is also observed in patients with other pancreatic diseases such as chronic pansteatitis and autoimmune pancreatitis (AIP) [9]. However, since chronic pancreatitis and AIP sometimes cause MPD stenosis, we cannot diagnose the primary disease as PC by this finding alone. Therefore, in the clinical situation in which the cause of stenosis is unclear, it is often difficult to judge whether this is due to PC. As for parenchymal abnormalities, upstream pancreatic parenchymal atrophy (PPA) arising from the site of MPD stenosis is often detected in patients with PC [10]. In addition to upstream PPA, partial PPA is reported to be suggestive of the presence of small PC including carcinoma in situ (CIS) in previous studies [2,11,12]. Thus, pancreatic parenchymal changes as well as localized MPD stenosis may provide a cue for the diagnosis of early PC.

It is now generally accepted that the detection of small PC ≤ 10 mm including CIS is indispensable to improve the prognosis of PC. However, there are no reports regarding the characteristic CT findings in patients with small PC (≤10 mm) [6,7,8,9,13,14,15,16,17]. In this study, we attempted to identify the specific CT findings of small PC by focusing on abnormalities in MPD and pancreatic parenchyma in patients with small PC including CIS.

## 2. Materials and Methods

### 2.1. Patient Selection

#### 2.1.1. Small PC Group

Patients with small PC were identified if they met both of the following criteria: (1) a pathological diagnosis of PC based on the analysis of a surgical specimen and (2) a tumor with size of ≤10 mm, including that of CIS. Patients were excluded if they met any of the following criteria: (1) diagnosis of intraductal mucinous papillary carcinoma based on the pathological analysis, (2) preoperative chemotherapy and/or radiation therapy for PC, or (3) a lack of contrast-enhanced CT (CE-CT) images. The tumor size was measured by pathological examination using surgically operated specimens. This research was approved by the Review Boards of Kindai University Faculty of medicine (Code: 31-136; Approval date: 17 October 2019).

#### 2.1.2. Benign MPD Stenosis Group

Patients were enrolled if they met all of the following criteria: (1) presence of localized MPD stenosis without detection of a mass lesion including pancreatic cysts causing MPD compression and/or MPD stenosis on abdominal ultrasonography, CE-CT, MRI, and EUS and (2) a benign stenosis based on the pathological analysis of a surgical specimen, no change in the MPD morphology for more than one year in follow-up imaging studies and negative for pancreatic juice cytology, or no change in the MPD morphology for more than five years in the absence of pancreatic juice cytology examination. Patients were excluded if they met any of the following criteria: (1) MPD stenosis due to pancreatic stones, (2) diagnosis of AIP, or (3) lack of CE-CT images.

### 2.2. Imaging Technique

All CE-CTs were scanned by a helical multidetector CT scanners (4, 16, 64, and 128 slice) at our institution. The scanning protocol included unenhanced and contrast material–enhanced biphasic imaging, the latter of which was consisted of arterial and portal venous phases after intravenous administration of 150 mL of ioversol (Optiray 320; Guerbet, Villepinte, France) or 150 mL of Iohexol: Omnipaque 350, General Electric Healthcare, Princeton, NJ, USA) at a rate of 3–5 mL/sec by using an automated power injector. Images were reconstructed at 5.0-mm thickness in the axial plane for unenhanced images and at 1.25–3.0-mm thickness in the axial and coronal planes for arterial and portal venous phase images. Most CT scans included both thin (1.25 mm) and thick (5 mm) image series. Volume-rendered and maximal intensity projection reconstructed images of arterial and venous structures were routinely generated by gastroenterologist or radiologists and were sent to the picture archiving and communication system for interpretation by radiologists. Arterial phase imaging was initiated 10 s after enhancement of the descending aorta to 100 HU, as measured using bolus-tracking methods (Bolus tracking; Toshiba or Snapshot pulse; GE Healthcare). Portal venous phase imaging was performed with a fixed scan delay of 72 s. Scan parameters included beam collimation of 128 × 0.625 mm (for GE scanners) or 64 × 0.625 mm (for Toshiba scanners), table rotation time of 0.5–0.6 s, spiral pitch of 1, a field of view to fit, 120 kVp (for GE scanners /Toshiba scanners), and 200–440 mA with use of automatic exposure control (maximum effective dose of 220 mAs or AutomA and SmartmA; GE Healthcare; noise index of 11). Images were reconstructed using iterative reconstruction algorithms (ASIR 20%; GE Healthcare).

### 2.3. Outcome Measurements and Definitions

The purpose of the present study is to evaluate the differential CT findings between the small PC group and the benign MPD stenosis group in a retrospective manner. CT images were assessed independently and blindly by two radiologists (M.T. and I.N.) with 24 and 8 years of CT interpretation experience, respectively. They were provided with an individual list of accession numbers that had been randomized using a random number generator (Microsoft Excel for Mac, version 16.16.11, Microsoft Corp., Redmond, WA, USA). Images were retrieved through the Picture Archiving and Communication System using the image accession number to prevent automatic retrieval of prior comparisons.

### 2.4. CT Findings

The following nine CT findings were assessed based on previous imaging studies [6,7,8,9,14,16]: presence of partial PPA, upstream PPA originating from the site of MPD stenosis, MPD enhancement corresponding to the stenosis area, MPD abrupt stenosis, attenuation difference, loss of fatty marbling, calcification adjacent to MPD stenosis, cystic lesion adjacent to MPD stenosis and tumor lesion. The reviewers assessed whether these findings were present in the small PC and/or benign MPD stenosis groups. Illustration and Definition of partial PPA and upstream PPA is shown Figure 1 and Figure 2. Partial PPA was defined when parameters obtained in CT met the following three criteria. First, a partial invagination of the pancreas parenchyma exhibited 4 mm or less in the width from the MPD wall (W1). Second, the length of partial PPA ranged from 10 to 25 mm (L1). Third, the width of upstream side of parenchyma was more than 6 mm (W2) (Figure 1A and Figure 2A). Upstream PPA was defined as an overall upstream parenchymal atrophy with 4 mm or less in the width of stenosis site (W1) and upstream side (W3) of parenchyma (Figure 1B and Figure 2B). These partial and upstream PPA were defined as atrophic morphology assessed by imaging and pathologic examinations. Additionally, these PPAs were regarded as independent atrophic morphology. MPD enhancement was defined as an MPD ductal wall enhancement that was greater than that of the adjacent pancreatic parenchyma [9]. Abrupt stenosis was defined as an abrupt rather than smooth stenosis of the MPD originating from the site of MPD dilation [9]. Attenuation difference of pancreatic parenchyma was defined as a difference in attenuation between the downstream and upstream sides of MPD stenosis [9]. Loss of fatty marbling was defined when the border between intra- or peripancreatic lipomatous tissue and pancreatic parenchyma was obscured [16]. The findings were calculated using a two-point scale; a finding was considered to be present when two reviewers agreed on its presence, and any disagreement was resolved by the opinion of the third reviewer. The relative frequency of specific imaging findings in each group was determined. Interobserver agreement for all signs was calculated using the weighted kappa statistic. Kappa values were interpreted as follows: < 0.00, poor agreement; 0.00–0.20, slight agreement; 0.21–0.40, fair agreement; 0.41–0.60, moderate agreement; 0.61–0.80, substantial agreement; and 0.81–1.00, almost perfect agreement.

### 2.5. Statistical Analysis

The differences in CT findings between the small PC group and benign MPD stenosis group were evaluated using univariate statistical tests; the Fisher exact test was used for categorical variables, while the t test was used for continuous variables. Interobserver agreement was assessed using k statistics. Two-tailed *p* < 0.05 was considered to indicate statistical significance, and the 95% CI was reported for each variable. Statistical analysis was performed using R software version 3.4.1 (The R Foundation for Statistical Computing, Vienna, Austria).

## 3. Results

### 3.1. Patient Selections

A total of 997 patients were diagnosed as PC at our institution between January 2004 and February 2020 through their medical records, and 33 patients had malignant tumors ≤ 10 mm. Nine of these 33 patients were excluded because the tumor size was initially larger than 10 mm, but the size was finally reduced 10 mm or less in the surgical specimen due to neoadjuvant chemotherapy. A total of 103 patients were diagnosed as localized MPD stenosis not associated with PC. Patients were excluded from the benign MPD stenosis group due to the following reasons; 38 patients with MPD stenosis due to pancreatic stones caused by chronic pancreatitis, 19 patients with isolated MPD stenosis due to AIP, nine patients with their follow-up period less than five years without pathological evidence for benign lesions, seven patients without CE-CT imaging, and two patients with their follow-up period less than one year despite negative result of pancreatic juice cytology. Ultimately, the final analysis included 24 patients in the small PC group and 28 patients in the benign MPD stenosis group (Figure 3 and Figure 4).

### 3.2. Patient Characteristics

The characteristics of the patients in the small PC group and the benign MPD stenosis group are shown in Table 1. There were no significant differences between the two groups. No significant difference was seen in PC-related clinical parameters such as the presence of chronic pancreatitis or diabetes, smoking habits, or alcohol consumption. This is probably due to the fact that the number of both groups is small. The diagnostic confirmation was obtained by pathological examinations on surgically operated specimens in all 24 patients in the small PC group. The tumor stage in small PC group was stage 0 (CIS) in 11 and stage Ia in 13 (tumor size; 1–5 mm in three and 6–10 mm in 10). Benign MPD stenosis was diagnosed by surgery in six patients. Fifteen cases with negative pancreatic juice cytology exhibited no changes at follow-up imaging studies for more than one year. Seven cases exhibiting no changes at follow-up imaging studies for more than five years were also included in this group.

### 3.3. Frequency of CT Findings

The frequency of CT findings in the two groups is shown in Figure 5. We assessed the presence of the following CT findings based on previous reports [6,7,8,9,14,16]; partial PPA, upstream PPA, MPD enhancement corresponding to the stenosis area, MPD abrupt stenosis, attenuation difference, loss of fatty marbling, calcification adjacent to MPD stenosis, cystic lesion adjacent to MPD stenosis and tumor lesion. As shown in Figure 5, the proportion of patients exhibiting partial PPA (45.8% vs. 7.1%, *p* < 0.01) and upstream PPA (33.3% vs. 3.6%, *p* = 0.01) was significantly higher in the small PC group than in the benign MPD stenosis group. The proportion of overall PPA (partial and upstream) was markedly higher in the small PC group than in the benign MPD stenosis group (79.2% vs. 10.7%, *p* < 0.01). Moreover, the proportion of abrupt MPD stenosis was also significantly higher in the small PC group than in benign MPD stenosis group (45.8% vs. 14.3%, *p* = 0.03).

In the assessment of the type of PPA in the small PC group, seven of the 13 patients with partial PPA were found to be CIS (63.6%) and four were found to be non-CIS (36.3%). Two of the eight patients with upstream PPA were CIS (25.0%), while six were non-CIS (75.0%). In this assessment, the proportion of the types of PPA (partial vs. upstream PPA) is not significantly different between patients with CIS and non-CIS (*p* = 0.17). Tumor lesion was detected in only three patients (12.5%) with the small PC group by blind reading, and these tumor diameters were 5 mm, 8 mm, and 9 mm, respectively. These data strongly suggest that the presence of partial or upstream PPA as indirect findings might be one of the diagnostic CT findings, suggesting the presence of small PC. Typical cases with small PCs exhibiting partial PPA (Figure 6) or upstream PPA (Figure 7) are shown. Figure 8 illustrates a typical case with benign MPD stenosis, which was not accompanied by PPA.

### 3.4. Interobserver Agreement

The results of interobserver agreement for each CT finding are shown in Table 2. A two-point scale revealed that substantial and excellent agreement regarding the detection of almost findings. Kappa values of partial PPA, upstream PPA and MPD with abrupt stenosis were 0.74, 0.68, and 0.57, respectively.

## 4. Discussion

In the present study, we assessed the CT findings associated with small PC with a diameter ≤ 10 mm including CIS in comparison to those with benign MPD stenosis. Here, we provide evidence that CT findings specific for small PC were partial PPA, upstream PPA, and/or MPD abrupt stenosis.

Diagnostic imaging modalities such as CT, MRI, and EUS are indispensable for the detection of PCs. However, the rates of direct tumor detection on such imaging analysis are not sufficient in patients with small PCs as defined by tumor size ≤ 20 mm; direct detection is achieved via abdominal ultrasonography in 52.6%, CT in 51.5%, and MRI in 45.1% [2]. In the present study focusing on small PCs ≤ 10 mm, the proportion of direct detection by blind reading of CT images was only 12.5% (3/24) in the small PC group. This very low rate for direct detection supports the idea that pick-up of the indirect imaging findings, highly suggestive of small PC, is important [3]. Several studies have assessed the utility of such indirect findings of PC in CT image [6,7,8,9]. Kim et al. [9] reported that the significant indicators of PC versus benign MPD stenosis obtained in CT images were long MPD stenosis (≥ 6.1 mm), abrupt MPD stenosis, non-MPD penetrating sign, attenuating difference, and CBD and MPD wall enhancement. Zaheer et al. [8] reported that the significant indicators of PC rather than AIP were the presence of a focal mass, PPA, non-diffuse parenchymal enlargement, non-diffuse parenchymal hypo-enhancement, and non-pancreatic halo or rim. As for the proportions of PPA in PC patients, those ranged between 13.3–53.0% [6,7,8,9] and were thus highly variable. In this study, the proportion of small PC patients exhibiting PPA was very high (79.2 %). This discrepancy can be explained by the definition of PPA and the size of PCs enrolled in this study. Regarding the definition of PPA, the presence of upstream PPA but not partial PPA was evaluated in the previous studies [6,7,8,9]. Furthermore, different from these previous studies focusing on PCs with all tumor sizes [6,8,9] or less than 20 mm [7], the current study has focused on patients with small PC ≤ 10 mm in size. Given that the mean diameter of the pancreatic parenchyma is around 20 mm, it might be impossible to point out partial PPA in PC patients bearing tumors more than 20 mm. In other words, the assessment of partial PPA may be a useful indicator only in the diagnosis of small PC ≤ 10 mm in size. Although upstream PPA has been considered to be highly associated with the presence of PCs [18], partial PPA of pancreatic parenchyma are reported to be suggestive of the presence of PC in recent studies and case report [2,11,12]. Kanno et al. reported that partial PPA on CT image was seen in 41.8% of patients with small PC (≤20 mm) [11] and in 42.0% of patients with stage 0 PC [2]. Satoh et al. [12] also reported that fibrosis and atrophy occur simultaneously with fatty changes on pathological assessments, and these changes are visualized as local parenchymal atrophy (partial PPA) on CT images. Here, we provide evidence that the assessment not only of upstream PPA but also of partial PPA might improve the detection rate of a small PCs.

As shown above, PPA might be associated with the presence of PC, but not chronic fibroinflammatory responses of the pancreas [19]. In this regard, PPA is considered to occur from the stage of low-grade PanIN, the precancerous stage of PC [20]. PC induces thinning of the acinar cell layer via disruption of flow in pancreatic duct. The thinning of acinar cell layer is followed by apoptosis and/or necrosis-mediated cell death. Loss of acinar cells is replaced by fibrotic changes and infiltration of immune cells and then visualized as PPA [21,22]. Thus, loss of acinar cell architecture accompanied by PPA is verified in the human pancreatic tissue bearing PC. These parenchymal changes are considered to be visualized as PPA on CT [2,12,23]. Thus, previous pathological studies support the idea that PPA is one of the characteristic pathological findings accompanied by PC. Therefore, PPA identified as CT indicators in this study might arise from pathological changes associated with PC, even its very small size. Considering these mechanisms, PPA may occur in the pancreas tissues surrounding the malignant tumors and then spread to the upstream parenchyma due to disruption of flow in pancreatic duct. If this is the case, partial PPA in patients with CIS may be detected at a higher incidence than non-CIS. However, in the present study, the proportion of the types of PPA (partial PPA vs. upstream PPA) was comparable between the patients with CIS and non-CIS (*p* = 0.17). We consider this point needs to be assessed in a larger number of cases.

Previous reports tried to identify and characterize CT findings in patients who had undergone abdominal CT before the clinical diagnosis of PC [13,14,16,17]. Gangi et al. [13] reported that MPD dilation and cutoff were the early findings of PC before the diagnosis (average time interval: 13.4 months). Another report by Ahn et al. [14] revealed focal hypoattenuation and upstream PPA were the independent predictors of PC (average time interval: 13.7 months). Two reports addressed CT findings in patients who received CT for other diseases before the detection of PC [16,17] and clarified the importance of small pancreatic nodules, MPD dilation, MPD interruption, and loss of fatty marbling as predictors of PC. Given the fact that pancreatic tumors were diagnosed at the advanced stage in these studies, identification and establishment of new CT findings could increase the detection rate for early PCs. In the present study, we tried to pick up specific CT findings in the patients with PC lesions ≤ 10 mm including CIS. In line with these previous studies [13,14,16,17], the proportion of patients exhibiting MPD abrupt stenosis was significantly higher in small PC group than in benign MPD stenosis group. Therefore, MPD abrupt stenosis as well as PPA may be useful for the diagnosis of small PC. Kim et al. [9] reported that the incidence of MPD abrupt narrowing was more frequently seen in patients with malignant MPD stenosis as compared with benign stenosis. They assumed that this may be attributed to the fact that PC causes more complete obstruction of the MPD than benign diseases [9]. These previous reports, together with our study, suggest that MPD abrupt narrowing and/or dilation is one of the common CT findings suggestive of PC irrespective of its size. While partial PPA was not assessed in the previous studies [13,14,16,17], it would be interesting to examine the frequency of partial PPA in the patients enrolled in these studies.

In clinical situations, localized MPD stenosis with or without upstream dilation is the most suspicious indicator of PCs [10]. This is also the case for small PC as shown by a report by Kanno et al. [2], in which MPD dilation was seen in 79.6% (72.0% in stage 0 and 82.2% in stage I patients) of patients with early PCs. However, the clinical significance of MPD abnormalities, which is often seen in CT images of PC patients, has not been fully clarified. Since localized MPD stenosis can be detected in both benign and malignant pancreatic disorders, we tried to extract CT findingsthat are specific for small PC and not present in benign MPD stenosis. Here we provide evidence regarding specific CT findings that are useful for identification of etiologies causing MPD stenosis.

As for patient exclusion criteria for benign MPD stenosis in our study, patients with MPD stenosis due to pancreatic stone, AIP and no change in the MPD morphology for less than five years without pathological analysis were excluded in the benign MPD stenosis group. Thus, patients exhibiting MPD stenosis without detection of a mass lesion including pancreatic cysts causing MPD compression or stenosis were enrolled to extract CT findings highly suggestive of small PCs, but not benign MPD stenosis due to unknown etiology. We recognize that the exclusion criteria in the present study might be strict as compared with previous studies [8,16]. In the previous study conducted by Kim et al. [9], the patient inclusion criteria were similarly strict to ours. MPD stenosis exhibiting no change in MPD morphology for five years was regarded as benign stenosis in this study. Regarding this time window for five years, Yachida et al. [24] reported the duration between the birth of cancer cells and metastasis could be estimated an average of 6.8 ± 3.4 years. Moreover, Yamao et al. [25] and Kuruma et al. [26] reported that CIS developed in patients without morphology change of MPD for a long time (2.1 years and 4.0 years, respectively). These previous studies support the view that our exclusion criteria for time window in pathologically-unproven benign MPD stenosis could be valid. In our knowledge, the present study is the first one that identified specific CT findings associated with very small PC ≤ 10 mm in comparison to those with benign MPD stenosis.

Two diagnostic approaches are available for PC, EUS-fine needle aspiration (FNA), and Serial pancreatic juice aspiration cytologic examination (SPACE). EUS is powerful detection tool for PC ≤ 20 mm in size [4,5] and specific EUS findings in patients with small PCs was recently reported [27,28]. Additionally, EUS-FNA has been widely used for pathological confirmation due to high sensitivity and specificity [29,30,31]. EUS-FNA is also useful for the pancreatic tumor ≤ 10 mm in size [32,33,34]. One of the disadvantages of EUS-FNA is that this procedure cannot be used for CIS due to the lack of visualization of tumor. SPACE, in which a nasopancreatic tube is placed by endoscopic retrograde cholangiopancreatography, is an alternative diagnostic procedure for especially CIS. SPACE has a high sensitivity and specificity for the detection of small PC or CIS [12,35]. Here we provide evidence that PPA and MPD abrupt stenosis are CT findings highly suggestive of the presence of small PC. Therefore, it is expected that such CT findings, in combination with EUS-FNA and/or SPACE, enable to diagnose very early stage PC.

Our study has the following strengths. First, we assessed the findings of small tumors ≤ 10 mm, including CIS. There was no previous study that assessed the CT findings related to such small-sized PC tumors in comparison to those with benign MPD stenosis. To improve the prognosis of PC, it is essential to diagnose patients with PC lesions ≤ 10 mm, and this may be done via the assessment of PPA. Second, we focused on partial PPA, which was recently reported as a new CT finding in small PC and we demonstrated that this finding is indeed a powerful indicator for small PC. However, our study has several limitations. First, enrolled patients are limited by strict patient criteria and the sample size is small. Verification of our findings requires future studies addressing the presence or absence of partial PPA, upstream PPA, and MPD abrupt stenosis in CT in a large number of small PC patients. Thus, it might be too early to define these findings as strong indicators of small PCs ≤ 10 mm. Second, the study was done in a retrospective manner; various CT scanners with various acquisition protocols (such as scan parameters and CT phases) were used. Third, the assessment of partial PPA in patients with pancreatic head stenosis was difficult due to the swelling of the pancreas head even in healthy subjects. Therefore, application of the definition for partial PPA into pancreatic head stenosis is difficult. Although our study highlights the importance of detection not only of upstream PPA but also of partial PPA, adequacy of parameters used for the definition of PPA need to be examined in future studies. Fourth, our study evaluated CT findings alone. MRI is also a powerful diagnostic tool for the detection of PC. Identification and establishment of MRI findings specific to small PCs might be useful to increase the detection rate of small PCs. It should be noted, however, that CT, a primary imaging modality for the diagnosis of pancreatic diseases, is more widely used for the detection of PC than MRI. Thus, the establishment of small PC-specific CT findings takes precedence over that of MRI findings. Fifth and finally, histologic confirmation could not be achieved in seven (25.0%) patients in the benign MPD stenosis group, although such patients exhibited no evidence of malignancy up to more than five years of follow up.

## 5. Conclusions

The presence of partial PPA, upstream PPA, and/or MPD abrupt stenosis on CT were highly suggestive of small PC with a diameter ≤ 10 mm, including CIS. These CT findings might be useful in patients in whom it is difficult to judge whether the MPD stenosis arises from benign or PC. We need to bear in mind a possibility that small PC ≤ 10 mm, including CIS, may exist upon encounter with partial and upstream PPA and MPD abruption stenosis on CT images. EUS-FNA and/or SPACE is recommended for the final diagnosis of such patients. Moreover, patients exhibiting these findings require strict observation because of the risk for development of malignancy.

## Figures and Tables

**Figure 1 diagnostics-10-00445-f001:**
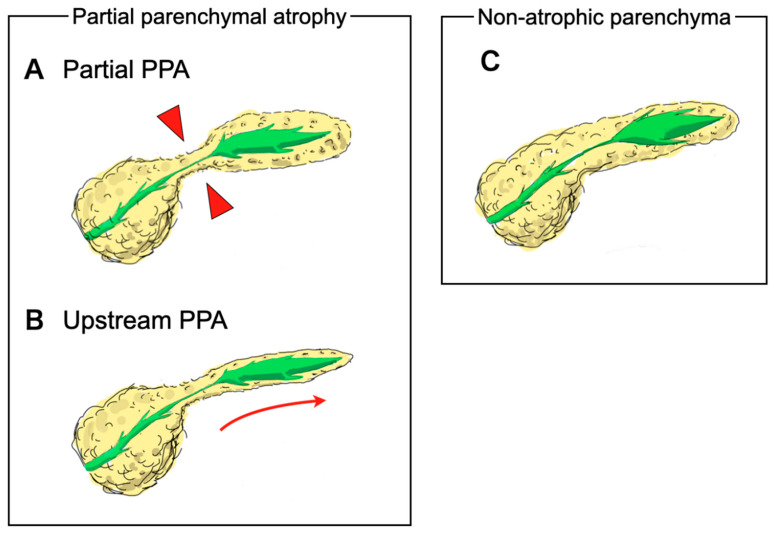
Illustrations of pancreatic parenchymal atrophy (PPA). PPA was categorized into partial PPA or upstream PPA. (**A**); Partial PPA was defined as an atrophic change corresponding to the distribution of main pancreatic duct (MPD) stenosis (red arrow heads). (**B**); Upstream PPA was defined as atrophic change of the whole upstream side of the parenchyma arising from the site of MPD stenosis (red arrow). (**C**); Non-atrophic parenchyma was defined as the absence of parenchymal atrophic change in the whole parenchyma despite the presence of main pancreatic duct stenosis.

**Figure 2 diagnostics-10-00445-f002:**
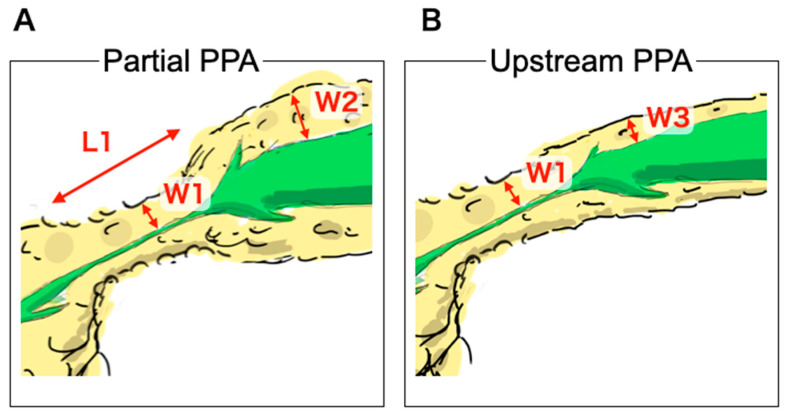
Definition of pancreatic parenchymal atrophy. (**A**); Partial pancreatic parenchymal atrophy (PPA) was defined when parameters obtained in CT met the following three criteria. First, a partial invagination of the pancreas parenchyma exhibited a width from the MPD wall of 4 mm or less (W1). Second, the length of the partial PPA ranged from 10 to 25 mm (L1). Third, the width of upstream side of parenchyma was more than 6 mm (W2). (**B**); Upstream PPA was defined as an overall upstream parenchymal atrophy with 4 mm or less in the width of stenosis site (W1) and upstream side (W3) of parenchyma.

**Figure 3 diagnostics-10-00445-f003:**
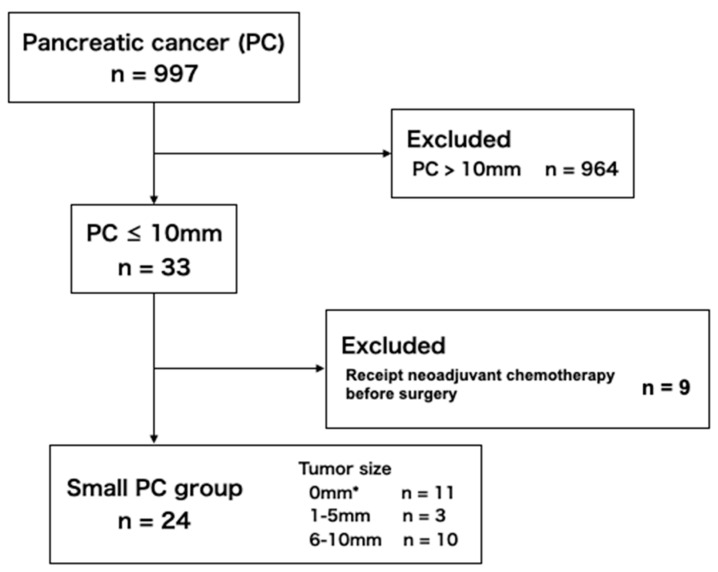
Flow-chart of the inclusion of patients in the small pancreatic cancer group. * Carcinoma in situ.

**Figure 4 diagnostics-10-00445-f004:**
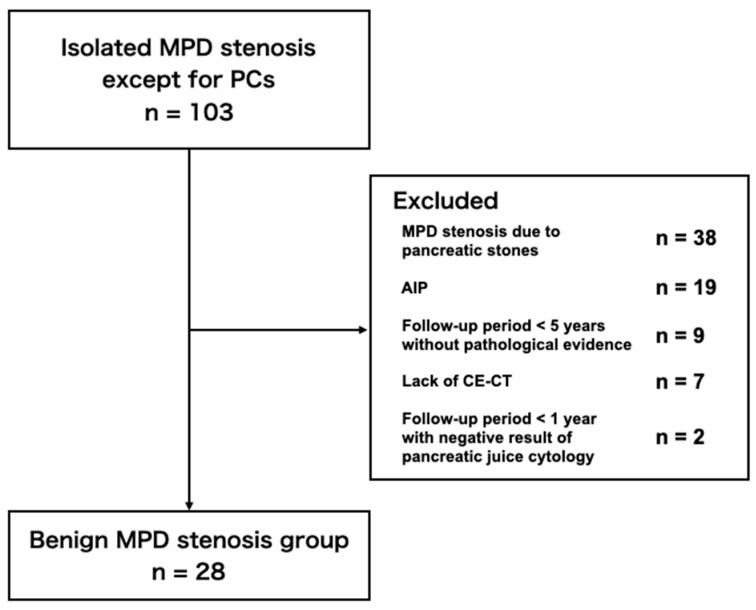
Flow-chart of the inclusion of patients in the benign main pancreatic duct stenosis group. AIP: autoimmune pancreatitis; CE-CT: contrast-enhanced computed tomography.

**Figure 5 diagnostics-10-00445-f005:**
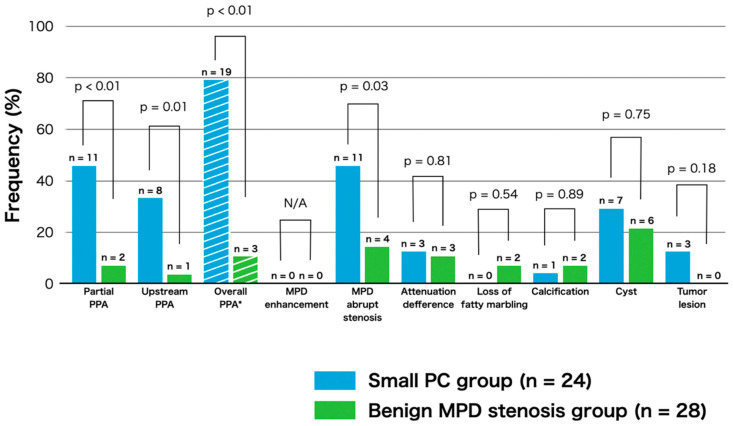
The proportions of patients exhibiting computed tomography findings in the small pancreatic cancer and benign main pancreatic stenosis groups. * Overall PPA was defined as the combination of partial and upstream PPA.

**Figure 6 diagnostics-10-00445-f006:**
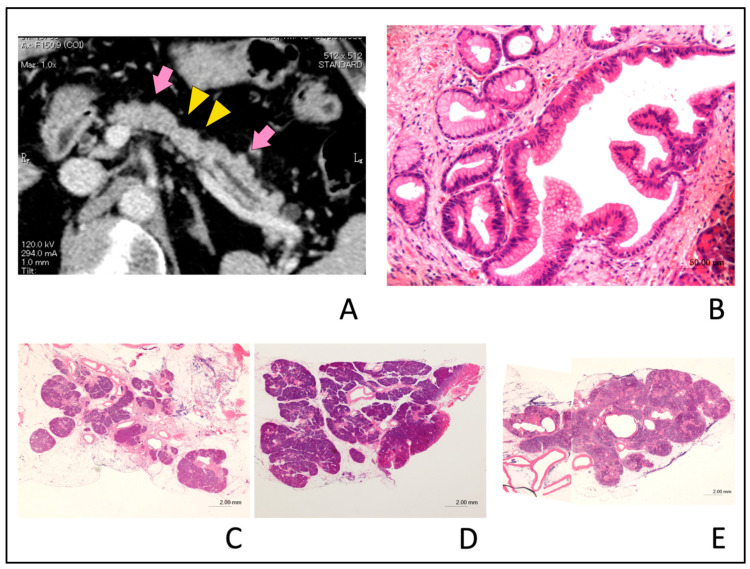
Typical computed tomography (CT) images showing partial pancreatic parenchymal atrophy (PPA) in an 81-year-old man with stage 0 (carcinoma in situ) pancreatic cancer. Localized main pancreatic duct (MPD) stenosis without tumor lesion in the pancreatic body was detected in CT for further examination of a small cyst in the pancreatic tail. (**A**); The area exhibiting PPA had an atrophic change corresponding to the distribution of MPD stenosis (yellow arrow head) and defined as localized atrophy compared with the upstream and downstream parenchyma (pink arrow). (**B**); Hematoxylin and eosin (H&E)-stained resected specimen. Histopathological examination showed high-grade pancreatic intraepithelial neoplasia (PanIN) of the main pancreatic duct (H&E stain, ×20 magnification). (**C**); Severe atrophy and fibrosis of the pancreatic parenchyma and focal fatty change adjacent to the high-grade PanIN (H&E stain, ×0.5 magnification). (**D**); Downstream side of pancreatic parenchyma had no atrophic change as compared with Figure 6C (H&E stain, ×0.5 magnification). (**E**); Upstream side of pancreatic parenchyma had slight fibrosis and focal fatty change as compared with Figure 6C. Acinar cell architecture was maintained (H&E stain, ×1 magnification).

**Figure 7 diagnostics-10-00445-f007:**
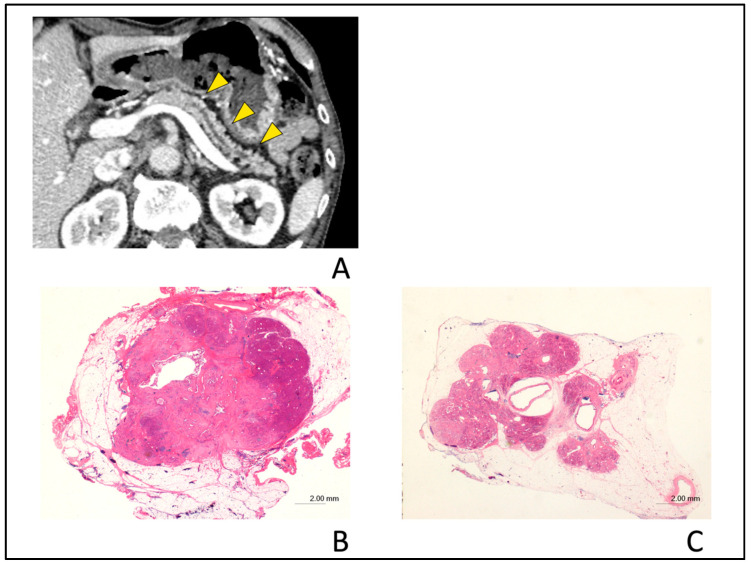
Typical computed tomography (CT) images showing upstream pancreatic parenchymal atrophy in a 72-year-old man with stage IA (tumor diameter of 10 mm in size) pancreatic cancer. Localized main pancreatic duct dilation was detected in the pancreatic body in the CT after the recurrence of acute pancreatitis and tumor lesion was detected in endoscopic ultrasonography. (**A**); Typical CT images showing upstream pancreatic parenchymal atrophy (PPA). The area exhibiting PPA has an atrophic change in the whole upstream side of the lesion (yellow arrow heads). (**B**); Hematoxylin and eosin (H&E)-stained resected specimen shows cancer nodule with a diameter of 10 mm in size (H&E stain, ×0.5 magnification). (**C**); Severe atrophy of pancreatic parenchyma and fatty change were observed in the upstream section from cancer nodule (H&E stain, ×0.5 magnification).

**Figure 8 diagnostics-10-00445-f008:**
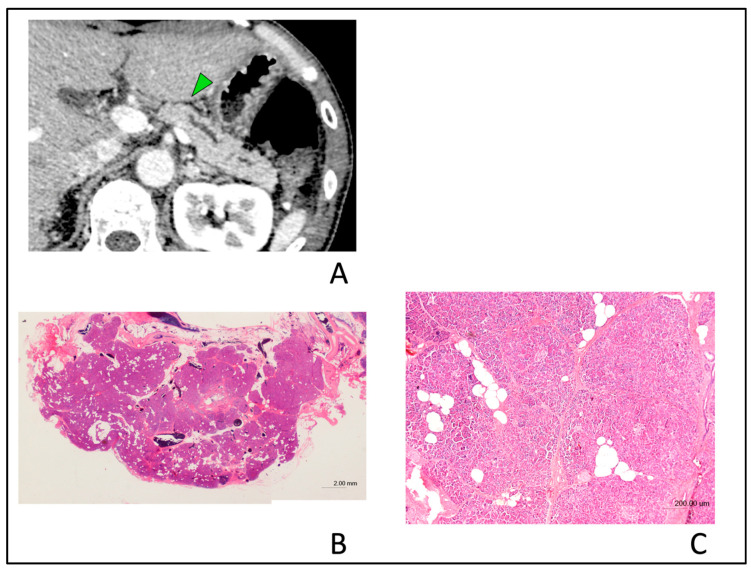
Typical computed tomography (CT) images showing non-atrophic parenchyma in a 55-year-old man with benign main pancreatic duct (MPD) stenosis. Localized MPD stenosis without tumor lesion was detected in the pancreatic body in CT after the recurrence of acute pancreatitis. (**A**); Typical CT images showing non-atrophic parenchyma. Non-atrophic parenchyma in a patient with chronic pancreatitis diagnosed in the resected specimen. CT image shows no evidence of pancreatic parenchymal atrophy regardless of the presence of isolated MPD stenosis (green arrow). (**B**); Hematoxylin and eosin (H&E)-stained resected specimen in the section of MPD stenosis area. Histopathological examination showed that acinar cell architecture was maintained (H&E stain, ×0.5 magnification). (**C**); Slight fibrosis and focal fatty change without parenchymal atrophy were observed in the pancreatic tissues surrounding stenosis MPD (H&E stain, ×4 magnification).

**Table 1 diagnostics-10-00445-t001:** Patient demographic and clinical characteristics.

	Small PC Group	Benign MPD Stenosis Group	*p* Value
	(n = 24)	(n = 28)	
Age in years, median (range)	73 (47–85)	69.5 (45–86)	0.15
Sex, n (%)			
Male	14 (58.3)	16 (57.1)	0.93
Female	10 (41.7)	12 (42.9)	
Diabetes mellitus, n (%)	6 (25.0)	6 (21.4)	0.98
Chronic pancreatitis *, n (%)	1 (4.2)	0 (0)	0.28
Body mass index ≥ 30 kg/m^2^, n (%)	0 (0)	1 (3.6)	0.94
Smoker (current or previous), n (%)	10 (41.7)	15 (53.6)	0.39
Heavy alcohol consumption **, n (%)	4 (16.7)	9 (32.1)	0.34
Main location of MPD stenosis, n (%)			
Pancreatic head	5 (20.1)	8 (28.6)	0.16
Pancreatic body	16 (66.7)	13 (46.4)	
Pancreatic tail	3 (12.5)	7 (25.0)	
Final confirmed detection method, n (%)			
Histological analysis	24 (100)	21 (75.0)	
Surgery	24 (100)	6 (21.4)	
Pancreatic juice cytology	0 (0)	15 (53.6)	
Follow up †	0 (0)	7 (25.0)	

PC: pancreatic cancer; MPD: main pancreatic duct; *: pancreatic stones present on CT image; **: ≥ three alcoholic drinks per day; †: Patients were monitored via clinical and radiologic examinations for more than five years.

**Table 2 diagnostics-10-00445-t002:** Kappa values for the two reviewers.

Findings	Kappa Value
Partial PPA	0.74
Upstream PPA	0.68
MPD enhanced with stenotic area	−0.02
MPD with abrupt stenosis	0.57
Attenuation difference	0.70
Loss of fatty marbling	0.65
Calcification	0.85
Cyst	0.88
Tumor lesion	0.64

PPA: pancreatic parenchymal atrophy; MPD: main pancreatic duct.

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
