# Peer review of "Partial Pancreatic Parenchymal Atrophy Is a New Specific Finding to Diagnose Small Pancreatic Cancer (≤10 mm) Including Carcinoma in Situ: Comparison with Localized Benign Main Pancreatic Duct Stenosis Patients"

_diagnostics, 2020, doi:10.3390/diagnostics10070445_

Round 1

Reviewer 1 Report

Dear Authors, I read with interest your paper which highlights the problem of undetected early pancreatic cancer on cross sectional imaging.

In this setting it is extremely important to become aware ofundirect signs signaling an early pancreatic cancer. Your paper appears well written in methods and language, and conclusions are well supported.

best regards

Author Response

Point 1: In this setting it is extremely important to become aware of undirect signs signaling an early pancreatic cancer. Your paper appears well written in methods and language, and conclusions are well supported.

 Response 1: We thank Reviewer 1 for these helpful comments.

Reviewer 2 Report

This study is to evaluate and identify the specific CT findings via focusing in main pancreatic duct and parenchyma in patients with small pancreatic cancer (smaller than 10mm in size). 24 patients with the small PC and 28 patients with the benign MPD stenosis were used and 9 CT findings were compared. The presence of PPA, for instance, was found to be in correlation with small PC existence. The study is important in terms of finding imaging markers that can be associated with small cancer. however, there are some concerns that authors may want to explain or clarify:

--the data size is very small, it is hard to generalize any findings from it. Any power calculation? any insights about finding other data sets, especially blind data set and use the same strategy to evaluate them ?

--in benign tumors for instance, MRI has been shown very successful in pancreas organ, why not authors utilize also MRI? is CT standard for small tumor in pancreas? I do not think there is an accepted screening strategy yet in pancreas cancer. CT is more common, but not sure about its distinction for different tumor types. Any insights? discussion? limitation?

--not sure how sensitive the system is, for instance, everything measured in the images, so PC when it was said it is 10mm, it can be 11mm measured but still closes to 10 or even 9 something. Did author think about putting a small margin round 10mm, and then see how results are changing? (patient numbers definitely change)

--in table 1, it seems no parameter seems to be correlating...that maybe due to size. Discussion should mention limitations of the work.

--why not include other cysts to make the database bigger and comparison more meaningful?

Author Response

Point 1: The data size is very small, it is hard to generalize any findings from it. Any power calculation? any insights about finding other data sets, especially blind data set and use the same strategy to evaluate them?

Response 1: We thank Reviewer 2 for the helpful comment. As the reviewer pointed out, the data size of our study is small. In addition, this study was assessed in a single-center and retrospective manner. This is probably because diagnosis of small pancreatic cancers (PCs) (tumor diameter ≤10 mm, including carcinoma in situ) is very difficult by established radiographic findings alone. In the clinical situation, such small pancreatic cancers (PCs) were very rare; we have experienced only 24 patients. In line with this idea, the proportion of small PC patients among total PC patients was only 2.4% in this study. This study was aimed to identify and establish radiographic findings specific to such small PCs in a limited number of patients. Considering such study backgrounds, we recognize that the data size is small and that our results need to be verified in the multi-center study with a large number of patients through the calculation of sample size. As mentioned above, we could not assess the power calculation due to a small number of PC cases. We added the limitation and weakness of this manuscript in the Discussion section of the revised manuscript in order to provide accurate information with the readers (Page 20, line 418).

 Point 2: In benign tumors for instance, MRI has been shown very successful in pancreas organ, why not authors utilize also MRI? is CT standard for small tumor in pancreas? I do not think there is an accepted screening strategy yet in pancreas cancer. CT is more common, but not sure about its distinction for different tumor types. Any insights? discussion? limitation?

Response 2: We thank Reviewer 2 for the helpful comment. As the reviewer mentioned, MRI is useful to evaluate the tumor character including solid or cyst, hyper- or hypo-vascular and containing fatty component or not. In addition, MRI is more useful to assess the main pancreatic duct (MPD) morphology. On the other hand, CT is superior to MRI in that the former modality can easily assess the pancreatic parenchymal morphology and vascularity by using thin image series. Thus, both CT and MRI may be useful for the detection of small PCs. It should be noted, however, that identification and establishment of CT rather than MRI findings specific to small PCs is necessary for improvement of the prognosis since the former modality can be widely used for the diagnosis of PCs. We added in the Discussion section of the revised manuscript explanation why we selected CT rather than MRI. (Page 20, line 427).

Point 3: Not sure how sensitive the system is, for instance, everything measured in the images, so PC when it was said it is 10mm, it can be 11mm measured but still closes to 10 or even 9 something. Did author think about putting a small margin round 10mm, and then see how results are changing? (patient numbers definitely change).

Response 3: We thank Reviewer 2 for the helpful comment. As the reviewer pointed out, the margin of PC was often unclear in the CT image and therefore the tumor size measured in CT might not be accurate. To avoid this problem, we evaluated the sizes of all tumors by pathological examination using surgically operated specimens. We clarified this point in Materials and methods section in the revised manuscript. (Page 2, line 83)

Point 4: In table 1, it seems no parameter seems to be correlating...that maybe due to size. Discussion should mention limitations of the work.

Response 4: We thank Reviewer 2 for the helpful comment. We presented the patient clinical data in Table 1; sex, age, risk factors of PC or pancreatic disease (diabetes, chronic pancreatitis, obesity, smoking and drinking) and the location of main pancreatic duct stenosis. We could not see any significant difference in these parameters between small PCs and benign MPD stenosis groups in this study. This may be attributed to a small sample size as this reviewer suggests. We referred to this in Result section in the revised manuscript. (Page 7, line 195)

Point 5: Why not include other cysts to make the database bigger and comparison more meaningful?

Response 5: We thank Reviewer 2 for the helpful comment. One aim of this study is to pick up specific CT findings highly suggestive of the presence of small PC in patients exhibiting MPD stenosis. As this reviewer points out, pancreatic cystic diseases sometimes cause MPD compression and stenosis. In this study, we tried to find out specific CT findings associated with small PCs in patients exhibiting MPD stenosis due to unknown etiology. Therefore, we excluded cases with pancreatic cystic diseases in this study and then tried to identify small PC-specific CT findings which are not influenced by the presence of pancreatic cysts. We clarified this point in the Material and methods section in revised manuscript and then as a result we reorganized the flow chart of benign MPD stenosis. (Page 2, line 94)

Reviewer 3 Report

A well-written study describes the use of partial and upstream pancreatic parenchymal atrophy, as well as abrupt stenosis of the main pancreatic duct detected by CT for diagnosis of pancreatic cancers smaller than 10mm.

Minor revisions:

Page 3, line 131 and 132 - add "L1" and "W2" in parentheses as well as

Add the same to the Figure 2 legend.

Figure 3 - there appears to be an asterisk next to Tumor size 0mm ("Small PC group"), please define the meaning of it or remove it.

Line 200 & 201 - since there is no mention of CEA and CA19-9 in the Table 1, please remove it.

Figure 5 - replace % above each column with the number of patients instead (y-axis already allows easy assessment of the %).

Figure 6C-E - are pictures with better resolution available? Alternatively, add higher magnification panels to it.

Figure 7A  - please define the yellow arrows in the figure legend

Figure 7B&C - are pictures with better resolution available? Alternatively, add higher magnification panels to it.

Line 313 and 314 - please rephrase the sentence.

Line 393 - should the word upstream be replaced by partial? Otherwise it appears upstream PPA is mentioned twice in the sentence.

Author Response

Point 1: Page 3, line 131 and 132 - add "L1" and "W2" in parentheses as well as

Add the same to the Figure 2 legend.

Response 1: We thank Reviewer 3 for this helpful comment. We added “L1” and “W2” in the text and the Figure 2 legend in the revised manuscript as suggested by the reviewer.

 Point 2: Figure 3 - there appears to be an asterisk next to Tumor size 0mm ("Small PC group"), please define the meaning of it or remove it.

Response 2: We thank Reviewer 3 for helpful comment. We added the explanation of asterisk in the Figure 3 legends of the revised manuscript as suggested by the reviewer.

  Point 3: Line 200 & 201 - since there is no mention of CEA and CA19-9 in the Table 1, please remove it.

Response 3: We thank Reviewer 3 for this helpful comment. We did not mention “CEA” and “CA19-9”, therefore, we removed these data from Table 1.

Point 4: Figure 5 - replace % above each column with the number of patients instead (y-axis already allows easy assessment of the %).

Response 4: We thank Reviewer 3 for helpful comment. We replaced the number above the column in the Figure 5 from the percent  to the number of patients as suggested by the reviewer.

Point 5: Figure 6C-E - are pictures with better resolution available? Alternatively, add higher magnification panels to it.

Response 5: We thank Reviewer 3 for helpful comment. We changed the pictures with high resolution (300 dpi) in the Figure 6 as suggested by the reviewer. 

Point 6: Figure 7A  - please define the yellow arrows in the figure legend

Figure 7B&C - are pictures with better resolution available? Alternatively, add higher magnification panels to it.

Response 6: We thank Reviewer 3 for helpful comment. We added the explanation of the yellow arrow head in the Figure 7A legends. Also, we changed the pictures with high resolution (300 dpi) in the Figure 7B and 7C as suggested by the reviewer. In addition, we changed the pictures with high resolution in the 8B and 8C.

Point 7: Line 313 and 314 - please rephrase the sentence.

Response 7: We thank Reviewer 3 for helpful comment. We rephrased the sentence to be understood easily as suggested by the reviewer.

Point 8: Line 393 - should the word upstream be replaced by partial? Otherwise it appears upstream PPA is mentioned twice in the sentence.

Response 8: We thank Reviewer 3 for helpful comment. We changed the sentence to “Although our study highlights the importance of detection not only of upstream PPA but also of partial PPA, …” as suggested by the reviewer.

Round 2

Reviewer 2 Report

questions are partially answered, I think the biggest limitation is that the results may not mean anything signifiant due to size limitation in the data.